# ClueVQA: Enhancing Query Based Retrieval in Video-LLMs with Answer Clues

## Abstract

Video-language models have achieved notable success in understanding complex visual narratives and answering fine-grained questions about video content. However, the computational burden of processing long videos - coupled with the growing size of modern models - restricts most approaches to processing only a limited number of frames. A widely adopted strategy to address this limitation is query-based frame retrieval, where frames are selected based on their semantic similarity to the given query. While effective in many cases, such methods are primarily limited to surface-level relevance matching and can fail when faced with implicit, ambiguous, or reasoning-intensive queries, potentially overlooking critical evidence in the video. In this work, we introduce **ClueVQA**, a novel retrieval framework that improves upon a standard query-based approach by generating and integrating supplementary answer clues and effectively utilizing them for frame selection. The answer clues are derived from the input query and a global scan of the video, which are then used to produce a secondary scoring distribution over frames. This clue-based distribution is then fused with the original query-based frame score distribution to yield a more informed frame selection. The final selected frames are passed to an off-the-shelf Video-LLM for answer generation. Extensive experiments on long-form VideoQA benchmarks, including MLVU, LongVideoBench, and VideoMME, show that our method considerably improves performance over a standard query-based retrieval method across different Video-LLMs.

## 1 Introduction

Recently, Vision-Language Models (VLMs) (Li et al., 2024; Bai et al., 2025b; Chen et al., 2024b), which were initially developed for image understanding, have been extended to general purpose Multimodal LLMs that can take as input images and videos. Variants of these models that are specifically adapted for video content, referred to as Video-LLMs (Yu et al., 2024; Wang et al., 2025), have demonstrated strong performance on video understanding tasks such as captioning and question answering, often rivaling their proprietary counterparts (OpenAI, 2025; Team et al., 2023). However, their application to long-form videos remains limited due to the substantial computational cost associated with processing many frames.

A commonly adopted solution is uniform sampling, where frames are selected at evenly spaced temporal intervals without regard to their semantic relevance to a particular task. While this method ensures broad temporal coverage, it is often inefficient as important frames may be missed, and redundant or irrelevant frames are frequently included. As a result, the model may fail to capture key events needed to answer a query, leading to hallucinations or incomplete reasoning in downstream tasks. Generally, in long video QA tasks (Zhou et al., 2024; Wu et al., 2024; Fu et al., 2024), only a small set of frames are more likely to contain information for providing an answer to a given query.

To address this, modern long-form VideoQA pipelines (Di et al., 2025; Ataallah et al., 2024; Qian et al., 2024; Shen et al., 2024) often use query-based frame retrieval. In this setup, a lightweight retrieval model computes the semantic similarity between each frame and the input question, and selects the top-k relevant frames to be passed to a Video-LLM for answer generation. This approach reduces computational overhead and works well for simple and information-rich queries (e.g., "What color is the car?" or "What does the sign say on the shop window?") wherein the answer is visually explicit and localized. However, many real-world questions are implicit, ambiguous, and reasoning-

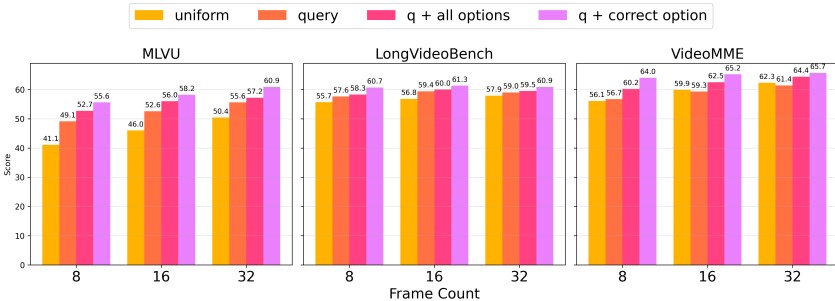

Figure 1: Performance comparison of different retrieval methods on long-form VideoQA benchmarks. (Best viewed in color and under zoom.)

intensive (e.g.,"What is the child likely looking at?" or "Why does the driver slow down before the intersection?"). These types of questions often depend on subtle causal or contextual cues that are poorly captured by surface-level query-frame similarity. In such cases, conventional retrieval can overlook key evidence - such as a pedestrian entering the crosswalk or a traffic light changing - which may result in degraded performance.

Given these limitations, a natural question arises: **Can retrieval performance be improved by incorporating additional answer-oriented clues, beyond what is directly available in the query?** To explore this, we conduct preliminary experiments on long-form VideoQA benchmarks such as MLVU (Zhou et al., 2024), LongVideoBench (Wu et al., 2024), and VideoMME (Fu et al., 2024), as illustrated in Figure 1. For each evaluation video, we uniformly sample up to 128 frames, and from these frames we select 8, 16 and 32 using the following methods: (1) uniform sampling, (2) query-based retrieval, (3) query concatenated with all multiple-choice answer options, and (4) query concatenated with only the correct answer option. It should be noted that a small number of subtasks refer to the answer choices directly in MLVU and LongVideoBench datasets. For those subtasks, we always keep the answer options appended to the query for fair comparison. We use SigLIP-SO400M/14 (Zhai et al., 2023) to extract frame and text features for creating frame score distribution (from which we sample top-k frames), and LLaVA-Video 7B model (Fu et al., 2024) to generate answers based on the selected frames. The results show that question-answering accuracy, used as a proxy for retrieval quality, improves significantly with the inclusion of answer options, achieving the highest performance when the correct answer is only attached to the query, indicating that auxiliary signals in the form of answer options can help find informative frames.

These answer options can be considered as explicit answer clues that, as demonstrated, significantly improve retrieval performance. However, in most real-life scenarios related to long-form VideoQA, we will not have access to them during inference. In such cases, the query-based retrieval might face limitations. To address these challenges, we propose **ClueVQA**, a retrieval-enhancement framework designed to improve frame selection by effectively generating and integrating complementary signals in the form of answer clues into the retrieval process for better frame selection.

To create answer clues in this framework, we construct a compression-based training scheme that allows a general-purpose small language model (SLM) with long-context modeling ability to be able to generate latent (implicit) answer clues and perform relevance-guided retrieval without explicit ground truth data for key frames containing answer clues. We train a small language model, referred to as *ClueSLM*, in two complementary modes: (1) a **compressor** that learns to represent each frame with a small set of summary tokens, and (2) a **clue generator** that selectively encodes video context based on the query to produce answer-relevant clues in latent space, acting as an information filtration module. During inference, we compute two frame relevance distributions: one based on the query and one based on the generated answer clues. These are then fused via a generalized noisy-OR mechanism to produce a robust and enhanced frame selection distribution. The top-ranked frames are then fed to any off-the-shelf Video-LLM for answer generation. Notably, our framework is modular and compatible with existing approaches: the query-based distribution can be derived from any retrieval scoring model or method, making ClueVQA a general and flexible enhancement to a broad class of retrieval-based VideoQA pipelines.

In summary, our contributions are as follows:

- We propose **ClueVQA**, a novel retrieval-enhancement framework that involves generating complementary signals in the form of answer clues derived from the input query and a global scan of the video, and integrating them into the query-based retrieval process for better frame selection.

- We introduce a general and efficient training scheme that enables clue generation and retrieval without requiring explicit ground truth for key frames containing answer clues. Our method converts a general-purpose SLM with long-context ability into a latent encoder working in dual modes - frame compression and clue generation - enabling it to understand local context and extract informative answer-relevant clues from long video content.

- We demonstrate considerable gains over the standard query-based retrieval baseline on multiple long-form VideoQA benchmarks such as MLVU, LongVideoBench, and VideoMME, while maintaining compatibility with existing Video-LLMs.

## 2 RELATED WORK

**Video-Language Models.** Research in multimodal learning has gradually evolved from image-based vision-language models to video-language models. Video-LLaMA (Zhang et al., 2023) incorporates visual and auditory modalities through dual Q-Formers and cross-modal pretraining. Video-ChatGPT (Maaz et al., 2023) applies spatial and temporal pooling to compress video features into compact tokens. PLLaVA (Xu et al., 2024) introduces a parameter-free adaptive pooling module that compresses visual tokens, enabling efficient processing of longer videos. LLaVA-Video (Zhang et al., 2024c) shows fine-grained video understanding through video-specific instruction tuning using a high-quality synthetic dataset, LLaVA-Video-178K, while Qwen2.5-VL (Bai et al., 2025b) handles images and videos with varying resolutions and frame rates through dynamic processing, and introduces absolute time encoding to enhance temporal reasoning and grounding.

**Key Frame Selection.** Most existing video-language models process all frames using a fixed token budget per frame, which leads to redundant visual information, higher memory usage, and increased computational cost. To address this, recent work has focused on adaptive frame selection based on the query. xu2023retrieval select question-relevant video chunks using cosine similarity. VideoStreaming (Qian et al., 2024) encodes each video clip into memory representations and selects the most relevant ones using query similarity. SALOVA (Kim et al., 2024) uses cross-attention between the query and frame-level segments to rank and select top-k candidates. Other approaches leverage large (multimodal) language models to perform frame scoring more holistically. For instance, VideoTree (Wang et al., 2024) uses an LLM to assign relevance scores to frame captions, SeViLA (Yu et al., 2023) scores frames one at a time using a vision-language model, whereas hu2025m and Frame-Voyager (Yu et al., 2024) employ a video-language model to score all frames simultaneously and select the top-k most relevant ones.

While these approaches improve efficiency and alignment with the query, they solely rely on semantic similarity between the query and frames. This limits their effectiveness in answering implicit, ambiguous, or reasoning-based queries, where important visual cues may be indirectly related to the question. As a result, key frames may be overlooked, leading to hallucinated or incomplete answers. In the next section, we introduce our framework for addressing these limitations by incorporating supplementary answer clues into the retrieval process.

## 3 METHOD

This section introduces how we designed and trained our module to generate implicit answer clues for enhancing query-based retrieval in long-form VideoQA. The overall ClueVQA framework with the proposed method is shown in Figure 2.

### 3.1 OVERVIEW: COMMON SAMPLING METHODS IN VIDEO LLMS

**Standard $k$-frame framework** A common strategy in Video-LLMs is the use of a $k$-frame framework, where a fixed number of frames are selected from the full video sequence for processing. Let $T$ denote the total number of frames in a video. A frame selection function $f_s$ is used to sample

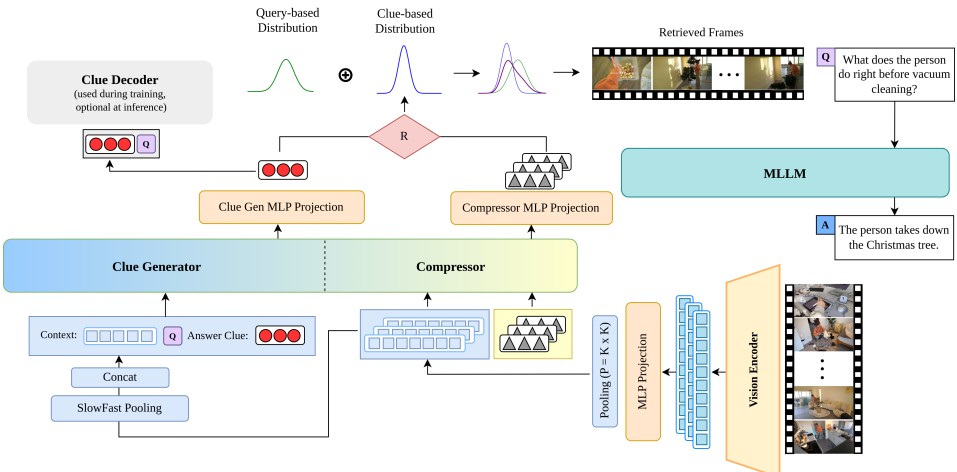

Figure 2: ClueVQA Framework. Video frames are encoded through the vision encoder, after which pooling is applied. The pooled features are passed to the clue generator and compressor. In the clue generator, they pass through additional SlowFast pooling layer, after which they are appended with a query (Q) and learnable "answer clue" tokens (*red circles*). In the compressor, each frame is appended with "summary" tokens (*gray triangles*) and processed independently. The processed answer clue and summary tokens are passed to the relevance score function (*R*) that determines the clue-based distribution, which is merged with an existing query-based distribution using generalized noisy-OR fusion strategy ($\oplus$). Top $k$ frames are retrieved from the new distribution and passed, with the query, to an off-the-shelf MLLM for answer generation. (Best viewed in color and under zoom.)

$k$ frames, denoted as $[x_1, x_2, \cdots, x_k]$, from the total $T$ frames, where each frame $x_i \in \mathbb{R}^{H \times W \times 3}$, with $H \times W$ representing the frame resolution, and typically $k \ll T$. Each sampled frame is passed through a pre-trained visual encoder $f_v$, and the resulting features are projected into the (multimodal) LLM's token space using an alignment module $g_a$. To reduce token dimensionality, a spatial pooling operation is usually applied:

$$h_i = Pooling(g_a(f_v(x_i))), h_i \in \mathbb{R}^{n \times d_{LLM}} \tag{1}$$

where $n$ is the number of tokens to represent a frame and $d_{LLM}$ is the hidden dimension of the LLM. Let $Q \in \mathbb{R}^{l \times d_{LLM}}$ denote the input embedding of the text question. The model then generates an answer $A$ by conditioning on the frame features and the question:

$$A = LLM(h_1, \cdots, h_k, Q) \tag{2}$$

**Frame sampling methods** Most existing Video-LLMs (Yu et al., 2024; Bai et al., 2025a; Wang et al., 2025) use uniform sampling method, where frames are extracted from a video at fixed, evenly spaced intervals. This ensures temporal coverage across the entire video, but can lead to inclusion of redundant or irrelevant frames and potential omission of key moments, especially in long videos.

In the case of long videos, most questions about a video can be answered using only a limited number of frames if we perform selection based on the particular task we are interested in: the given query about the video. A common way to perform such selection involves using vision-text encoder that can assign a semantic relevance score between the query and each frame:

$$f_s(Q, [x_1, \cdots, x_T]) =$$
$$\arg \operatorname{top}_k \left( f_r(x_1, Q), \cdots, f_r(x_T, Q) \right) \tag{3}$$

where $f_s \in \mathbb{R}^k$ is a function that returns a set of selected frame indices, $Q$ is the question (query), $x_i$ are video frames, and $f_r$ is a function that assigns relevance score between the given video frame and the question.

## 3.2 DESIGN & TRAINING PROCESS WITH *ClueSLM*

In Figure 1, we saw that the performance of query-based selection can be significantly improved if we include the answer options when performing the frame selection, which serve as answer clues and help the retrieval module select more informative frames.

We can more broadly define the answer clues in the context of VideoQA as the information that can be useful to answer the given question about a video. That information will exist inside the video as long as the given question is related to that video, and it can be extracted (and reflected in the answer) by a strong Video-LLM itself when it is given sufficient context about the video and the corresponding question as an input. However, when we are dealing with long videos, answer clues can often be sparsely located within a few frames inside the video. Thus, finding it becomes ineffective and inefficient, both due to the extremely large number of tokens, limited content length and extremely large parameter count in modern Video-LLMs.

Most RAG-based VideoQA pipelines (Xu et al., 2023; Wang et al., 2024; Qian et al., 2024; Hu et al., 2025) successfully decoupled the process of finding query-relevant frames and answer generation. We argue that identifying answer-relevant clues, which can often be complementary to query-relevant clues, can also be decoupled from direct question answering. Unlike direct answer generation, which requires fine-grained reasoning and intensive computation, clue generation focuses on identifying compressed, high-level semantic signals that are predictive of the answer, which can be a simpler task. Therefore, we propose to offload this process to a seperate compact module based on a general-purpose small language model (SLM) with long-context modeling ability.

This model, called ***ClueSLM***, operates in two complementary roles: as a **clue generator** $f_{cg}$, which produces latent answer clues, and as a **frame compressor** $f_{comp}$, which generates compact frame representations. These two functions share weights and operate within a common latent space, enabling both local (per-frame) compression and global (video-level) clue generation based on the given query. This shared representation allows us to later compute relevance scores between compressed frames and generated answer clues.

Initially, we sample $k$ frames from a video, and each sampled frame is passed through a pre-trained visual encoder $f_v$, and the resulting features are projected into the *ClueSLM*'s token space using an alignment module $g_a$ after applying light spatial pooling (2x2):

$$h_i = Pooling(g_a(f_v(x_i))), h_i \in \mathbb{R}^{n \times d} \tag{4}$$

where $n$ is the number of tokens to represent a frame and $d$ is the hidden dimension of the *ClueSLM*.

The clue generator is formulated as:

$$C' = f_{cg}(h'_1, \cdots, h'_k, Q, C),$$
$$A = Decoder(g_{cg}(C'), Q) \tag{5}$$

where $C$ and $C' \in \mathbb{R}^{m \times d}$ denote a set of $m$ learnable "answer clue" tokens appended to the end of the input and corresponding output tokens, respectively. $g_{cg}$ is an MLP projector between the clue generator and a decoder (pretrained MLLM shown as clue decoder in Figure 2) input space, whereas $h'_i$ represents transformed $h_i$ frame embeddings obtained using the SlowFast pooling strategy (Zhang et al., 2024c). In this approach, a stride hyperparameter controls which frames are processed with lighter spatial pooling (the original pooled features maintained) to preserve detail, and which are processed with heavier pooling (2x2 additional spatial pooling) to reduce token count. The resulting representations are interleaved to balance spatial fidelity and efficiency.

The compressor processes each frame independently as a batch, and it is defined as:

$$S'_i = f_{comp}(h_i, S(h_i)),$$
$$A = Decoder(g_{comp}(S'_1), \cdots, g_{comp}(S'_k), Q) \tag{6}$$

where $S(h) \in \mathbb{R}^{j \times d}$ denotes the function that adaptively pools frame features to create compressed $\sqrt{j} \times \sqrt{j}$ feature map (flattened to $j$ tokens with $d$ hidden dimension), which serve as initialization for the "summary" tokens for each frame, while $g_{comp}$ is the corresponding MLP projector for aligning compressor outputs with the decoder. $S'$ represents the output summary tokens, refined by $f_{comp}$. Unlike answer clue tokens, summary tokens are not learnable tokens, and the role of $ClueSLM$ is to

refine them based on more fine-grained frame embeddings, which are concatenated with those tokens in the compressor.

It should be noted that a pre-trained MLLM decoder is used to independently decode the output features from $f_{cg}$ and $f_{comp}$ to generate the answer $A$ to the given question. During training, we duplicate the batch where one copy will contain the summary tokens and another copy will contain the answer clue tokens. As the numbers of both types of tokens is small, duplicating the batch results only in marginal increase in computational cost, however in this way we can train the $ClueSLM$ to be able to function in dual modes. It should not be confused with an off-the-shelf "MLLM" used during inference, which is shown in Figure 2. The MLLM decoder is only used during training and is optional during inference, which can be helpful to explicitly understand what kind of information is represented inside the latent features of the clue generator and compressor. The native vision encoder of the MLLM decoder is used to extract frame-level features $h_i$, allowing *ClueSLM* to function as a new bridge between the vision encoder and the MLLM decoder.

We consider a three-stage training procedure to effectively train our *ClueSLM* in the role of a clue generator and compressor. In all stages, the vision encoder and MLLM decoder are frozen.

- **Stage 1: Modality Alignment.** We use 558K image-caption pairs from the CC3M dataset (Sharma et al., 2018), filtered by LLaVA (Liu et al., 2023). During this stage, an image is treated as a frame. We train learnable answer clue tokens and the projection layers $g_a$ and $g_{cg}$, while keeping all other components frozen. The compressor remains inactive in this stage.

- **Stage 2: Image-Instruction Tuning.** Using 763K image-instruction pairs from liu2024llavanext, we activate both the clue generator and compressor components of *ClueSLM*. We adopt the AnyRes (Li et al., 2024), where each image is divided into multiple blocks, processed independently by the vision encoder with the resized original image, all of which treated as frames (only spatial pooling applied with no SlowFast method). We introduce the $g_{comp}$ projection head and jointly train it alongside $g_a$, $g_{cg}$ and the answer clue tokens, all initialized from the Stage 1 checkpoint. This stage enables the model to process visual information and develop instruction-following ability in latent space.

- **Stage 3: Video-Instruction Tuning.** We further train *ClueSLM* on 400K video-instruction QA pairs drawn from five datasets: LLaVA-Video-178K (Zhang et al., 2024c), NeXT-QA (Xiao et al., 2021), ActivityNetQA (Yu et al., 2019), PerceptionTest (Pătrăucean et al., 2023), and LLaVA-Hound (Zhang et al., 2024b). These include captions, open-ended questions, and multiple-choice QA. All trainable modules are initialized from Stage 2. This stage enables the model to learn global video understanding and effective frame-level compression.

### 3.3 INFERENCE PROCESS WITH *ClueSLM*

After training, our *ClueSLM* operates in two modes: as a clue generator and as a compressor. The output tokens from both paths are projected into a shared latent space via the MLP projectors $g_{cg}$ (for answer clue tokens) and $g_{comp}$ (for compressed frame tokens), which are aligned during training. This alignment allows us to compute the semantic similarity between each frame's summary tokens and the generated answer clue tokens.

We use standard cosine similarity to calculate relevance score:

$$c_{r_i} = f_r(\hat{S}'_i, \hat{C}') = \frac{\hat{S}'_i \cdot \hat{C}'}{\|\hat{S}'_i\| \|\hat{C}'\|}, \quad i = 1, 2 \cdots, K \tag{7}$$

where $\hat{S}'_i$ is the mean of projected summary tokens for frame $i$, $\hat{C}'$ is the mean of the projected answer clue tokens, and $K$ is the total number of densely sampled frames (where $k \ll K \leq T$, much higher than $k$ frames used during training).

These relevance scores are then converted into a clue-based probability distribution using a softmax with temperature $\tau$:

$$P_c(i) = \frac{\exp(c_{r_i}/\tau)}{\sum_{j=0}^{K} \exp(c_{r_j}/\tau)}, \quad i = 1, 2 \cdots, K \tag{8}$$

When it comes to the scores representing the relevance between the frames and the query, namely $q_{r_i}$, it can be computed in multiple ways, but the most cost-effective method is using pretrained vision-text encoder features. Using this method, we generate the scores and convert them into a query-based probability distribution using softmax with the same temperature $\tau$ used above:

$$P_q(i) = \frac{\exp\left(q_{r_i}/\tau\right)}{\sum_{j=0}^{K} \exp\left(q_{r_j}/\tau\right)}, \quad i = 1, 2 \cdots, K \tag{9}$$

To combine both information sources, we merge $P_q$ and $P_c$ based on the following fusion strategy:

$$P_{q \oplus c}(i) = \mathcal{N}\left(1 - (1 - P_q(i))^{w_q}(1 - P_c(i))^{w_c}\right) \tag{10}$$

where $\mathcal{N}(\cdot)$ denotes normalization over the range (division by the sum across all entries). $P_{q \oplus c}$ is based on generalized noisy-OR (Pearl, 1988; Srinivas, 1993) fusion strategy, which captures the idea that either source alone can justify high relevance and it avoids underestimating the combined probability when both inputs are strong. The weights $w_q$ and $w_c$ allow control over the contribution of each source. For simplicity, we set $w_q = w$, where $w \in [0, 1]$, and $w_c = 1 - w$. In the Appendix (Table 5), we experiment with different merging methods such as arithmetic mean, max and harmonic mean, finding that weighted noisy-OR method performs the best.

During inference, after we generate the $P_{q \oplus c}$ distribution over a large set of $K$ densely sampled frames, we select a small subset of frames with highest scores. The selected frames are sorted in temporal order and then given to an off-the-shelf Video-LLM with the query for answer generation.

## 4 EXPERIMENTS

In this section, we first outline our experimental setup, including training and evaluation details. Then, we demonstrate that our framework improves the performance of several strong open-source Video-LLMs across multiple long-form VideoQA benchmarks, with additional analysis related to compression performance and latent answer clues. The details related to training, evaluation, compressor ablation and further analysis of our framework are given in the Appendix.

### 4.1 EXPERIMENTAL DETAILS

**Implementation Details**    We use LLaVA-OneVision 7B (Li et al., 2024) as the MLLM decoder during training. This model is chosen for its strong multimodal performance and simple architecture, which employs an MLP-based projector between the vision encoder and the language model, making it well-suited for our integration. We adopt the SigLIP-SO400M/14 (Zhai et al., 2023) vision encoder, which is also used natively in LLaVA-OneVision. The corresponding text encoder of the SigLIP model is used to compute query-based frame relevance scores during evaluation. We use 64 summary tokens and 64 answer clue tokens during both training and inference. During video-instruction tuning, up to 48 frames are uniformly sampled from each video.

For our *ClueSLM*, we use Rene (Cartesia, 2024), a compact and efficient 1.3B language model with long-context modeling ability, composed of alternating Mamba-2 (Dao & Gu, 2024) and MLP layers, with some sliding-window attention layers interspersed. We extract hidden states from the 36th layer (out of 48 total layers), which leaves around 0.8B parameters, striking a good balance between semantic abstraction and computational efficiency. This choice is further supported by prior work (Qian et al., 2024), which showed that when repurposing a language model as a latent encoder, intermediate layers - especially those not too deep - tend to yield stronger performance.

**Evaluation Benchmarks**    We perform evaluation on 3 widely used long-form VideoQA benchmarks: Video-MME (Fu et al., 2024), LongVideoBench (Wu et al., 2024), and MLVU (Zhou et al., 2024). These benchmarks include videos ranging from a few minutes to around two hours, thus testing the capabilities of Video-LLMs on long-form video understanding task. Following common practice (Zhang et al., 2024c), on VideoMME, we report the results of three subsets based on the video length (short: 30-120 seconds, medium: 4 - 15 minutes, long: 30 - 60 minutes).

Additionally, we analyze performance of $ClueSLM$ as a compressor on image-based QA benchmarks such as GQA (Hudson & Manning, 2019), VQAv2 (Antol et al., 2015), TextVQA (Singh et al., 2019),

and MMBench (Liu et al., 2024) to test the impact of stage 2 training. We also test the compression ability of $ClueSLM$ after video-instruction tuning stage on the above mentioned videoQA datasets and an additional dataset, called NextQA (Xiao et al., 2021), which consists of short videos averaging less than 2 minutes in length, focusing primarily on short-form question answering. For all expriments, we utilize the "lmms eval" library (Zhang et al., 2024a), which offers great flexibility for evaluating MLLMs and ensures reproducibility.

## 4.2 EXPERIMENTAL RESULTS

**VideoQA Performance** In Table 1, we experiment with three Video-LLMs, LLaVA-Video (Yu et al., 2024), InternVideo 2.5 (Wang et al., 2025), and Qwen 2.5-VL (Bai et al., 2025a) (all approximately 7B in size), under three sampling strategies: uniform sampling, query-only retrieval, and our clue-enhanced retrieval. We uniformly sample a large number of frames (128) and determine the query-based and clue-based distributions, after which we select top 8 and 16 highest scoring frames (out of 128 frames) based on the selection method. The optimal merge weights for $w_q$ and $w_c$ are 0.7 and 0.3, respectively, based on testing $ClueSLM$ with MLVU (Zhou et al., 2024) dev set. From the results, we can see that, across all three benchmarks, clue-enhanced improves performance compared to the query-only retrieval in the majority of the cases across all Video-LLMs. The performance tends to be consistently better in more challenging low-frame setting (8 frames) and tasks with a focus on long-form videos. In the Appendix, we additionally include the results with 32 frame setting (Table 6), ablate different configurations for merge weights (Figure 5) and also report results with a different vision-text encoder such as CLIP (Radford et al., 2021) (Table 3), all of which further support the effectiveness of our framework.

| Model | Frame Count | Video-MME (wo-subs) | | | LongVideoBench val | MLVU test |
|---|---|---|---|---|---|---|
| | | Short | Medium | Long | | |
| LLaVA-Video | $8_u$ | 67.7 | 53.7 | 46.9 | 55.7 | 41.1 |
| LLaVA-Video | $8_q$ | 66.8 | 54.1 | 49.1 | 57.6 | 49.1 |
| LLaVA-Video | $8_{q \oplus c}$ | **69.4** | **54.9** | **50.0** | **57.7** | **50.3** |
| LLaVA-Video | $16_u$ | 70.9 | **58.8** | 49.9 | 56.8 | 46.0 |
| LLaVA-Video | $16_q$ | **71.3** | 55.9 | 50.8 | 59.4 | 52.6 |
| LLaVA-Video | $16_{q \oplus c}$ | 71.1 | 56.4 | **52.0** | **59.8** | **54.9** |
| InternVideo2.5 | $8_u$ | 62.8 | 52.3 | 45.9 | 50.6 | 42.3 |
| InternVideo2.5 | $8_q$ | 64.4 | 53.1 | 47.7 | 55.4 | 47.8 |
| InternVideo2.5 | $8_{q \oplus c}$ | **64.6** | **53.4** | **48.2** | **55.5** | **48.3** |
| InternVideo2.5 | $16_u$ | 65.8 | 55.7 | 47.2 | 53.6 | 43.4 |
| InternVideo2.5 | $16_q$ | **68.0** | 54.8 | 48.0 | 56.3 | 51.9 |
| InternVideo2.5 | $16_{q \oplus c}$ | 67.7 | **56.0** | **49.1** | **57.2** | **53.4** |
| Qwen2.5-VL | $8_u$ | 62.1 | 51.2 | 47.0 | 52.5 | 37.0 |
| Qwen2.5-VL | $8_q$ | 63.2 | 52.8 | 47.9 | 55.8 | 44.5 |
| Qwen2.5-VL | $8_{q \oplus c}$ | **65.7** | **53.2** | **50.1** | **57.1** | **45.2** |
| Qwen2.5-VL | $16_u$ | 66.2 | 56.2 | 48.3 | 55.2 | 38.9 |
| Qwen2.5-VL | $16_q$ | 65.4 | 56.7 | 51.9 | 57.7 | **46.5** |
| Qwen2.5-VL | $16_{q \oplus c}$ | **68.3** | **56.8** | **52.2** | **58.0** | 46.3 |

Table 1: Performance on long-form VideoQA benchmarks. All 3 Video-LLMs are approximately 7B in size. Bolded score indicates the highest performance, underlined score indicates the second highest performance. The frame count subscripts represent the sampling method: $u$ - uniform sampling, $q$ - query-based selection, where $k$ highest scoring frames are selected based on query relevance, and $q \oplus c$ - based on our proposed framework, where $k$ highest scoring frames are selected based on relevance to the query and generated answer clues.

**Compression Analysis** In Table 2, we study the token-compression side of our framework by evaluating LLaVA-OV, used as an MLLM decoder during training, with a default adaptive pooling compression method, and with our learned compressor. The summary tokens are initialized from the adaptively pooled features, thus we can get a clear understanding of the compressor's role in modifying those features. Additionally, since summary tokens are not learnable tokens, we can try different token configurations during inference even if we trained with a fixed configuration by applying different adaptive pooling sizes. We can see that across most of the benchmarks and token settings, the compressor improves performance compared to the vanilla adaptive pooling setting.

Additionally, we test the compression ability of our clue generator with its corresponding projector, which only uses 64 "answer clue" tokens for the whole input, while in other settings, the total number of tokens in the input is considerably higher. We observe that even with a very restricted token setting, the clue generator can maintain high accuracy. This indicates that the clue generator can effectively extract the information needed to answer a question based on the video, serving as answer clues.

| Model | GQA | VQAv2 | Text VQA | MM Bench | Video MME | LongVideo Bench | MLVU | NextQA |
|---|---|---|---|---|---|---|---|---|
| | test | val-lite | val | en-dev | wo-subs | val | test | mc |
| LLaVA-OV$_D$ | 62.2 | 80.3 | 76.0 | 80.8 | 58.5 | 55.3 | 48.4 | 78.8 |
| LLaVA-OV$_{64}$ | 58.0 | 69.8 | 43.4 | 74.6 | 56.0 | 55.8 | 44.2 | 75.7 |
| LLaVA-OV$_{64}^{comp}$ | **60.4** | **72.4** | **60.4** | **74.8** | **57.6** | **56.4** | **46.0** | **78.0** |
| LLaVA-OV$_{16}$ | 49.8 | 56.8 | 20.3 | 65.1 | 51.3 | 50.9 | 40.3 | 72.9 |
| LLaVA-OV$_{16}^{comp}$ | **57.5** | **68.8** | **46.2** | **71.2** | **55.6** | **54.5** | **45.1** | **76.7** |
| LLaVA-OV$_4$ | 39.2 | 44.9 | 10.4 | 53.4 | 47.2 | **48.1** | 32.6 | 65.3 |
| LLaVA-OV$_4^{comp}$ | **48.3** | **54.9** | **24.8** | **61.1** | **51.1** | 47.6 | **39.3** | **71.0** |
| LLaVA-OV$_1$ | 35.0 | 39.8 | 8.8 | 40.6 | 44.4 | **44.4** | 29.2 | 58.4 |
| LLaVA-OV$_1^{comp}$ | **39.5** | **43.6** | **12.6** | **42.1** | **45.2** | 42.6 | **29.5** | **63.3** |
| LLaVA-OV$^{cg}$ | 57.9 | 69.8 | 47.4 | 73.5 | 51.6 | 48.8 | 36.9 | 75.4 |

Table 2: Compression performance with LLaVA-OV (7B MLLM decoder used during training). The subscripts under model name represent the number of tokens used to represent an image block or a video frame. $D$ - default encoding method was used. The models without superscript evaluated with adaptive pooling compression, while the models with *"comp"* and *"cg"* evaluated with the output tokens of our $ClueSLM$ in the role of *compressor* module and *clue generator* module, respectively.

**Answer Clue Analysis** To investigate the latent answer clues produced by *ClueSLM* in the role of a clue generator, we conduct t-SNE (van der Maaten & Hinton, 2008) cluster analysis. We use the VideoMME (Fu et al., 2024) dataset, which classifies videos into multiple categories. From this dataset, we extract a subset representing six unique categories (shown in Figure 3). We then query the clue generator with the prompt: "To which one of the categories does the video belong? (list all categories)". For each sample, we extract the generated latent answer clues and compute their average representation. In Figure 3, we observe that samples from the same category are located near each other, and well-formed clusters emerge. As some videos can be ambiguous and belong to multiple domains, we can also see that a few samples end up being represented in a different (but similar) domain. Overall, this analysis suggests that the latent answer clue tokens are adaptive and encode information specific to the video.

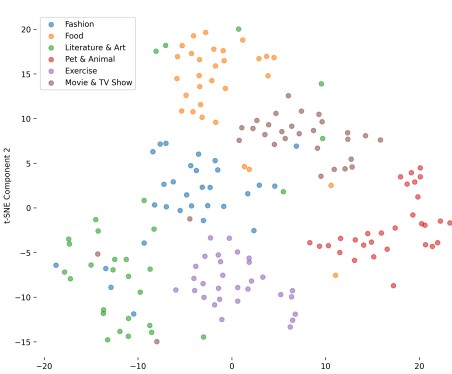

Figure 3: t-SNE analysis of answer clue latents on VideoMME (Fu et al., 2024) dataset.

## 5 CONCLUSION

We introduced **ClueVQA**, a framework designed to improve the standard query-based retrieval with complementary signals in the form of answer clues. We proposed a method that generates such clues implicitly using a general-purpose SLM trained to operate in both clue generation and frame compression modes, enabling relevance scoring in a shared latent space. Our framework is broadly applicable, and can be integrated with most Video-LLMs without architectural changes. Experiments on long-form VideoQA benchmarks demonstrate that ClueVQA delivers considerable performance gains across models and retrieval settings, offering a practical path for enhancing future Video-LLM systems.

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

# A  APPENDIX

## A.1  IMPLEMENTATION DETAILS

During the image-instruction tuning stage, we adopt the AnyRes (Li et al., 2024) scheme with bilinear interpolation. In this setup, each high-resolution input image is divided into multiple blocks, each of size $3 \times 384 \times 384$, and individually processed by the vision encoder, which produces a $27 \times 27$ feature map per block.

In the clue generator mode, we apply $2 \times 2$ bilinear pooling to each image block to reduce spatial resolution, and concatenate the resulting pooled blocks with the input query. We append 64 learnable answer clue tokens to the end of this sequence, which serve as the target for latent clue generation. In the compressor mode, each image block is bilinear-pooled to a $19 \times 19$ feature map to preserve finer visual detail while balancing efficiency. These feature maps are then processed independently. We generate 64 summary tokens per block by applying adaptive pooling to the $19 \times 19$ grid (resulting in an $8 \times 8$ token layout). The resulting summary tokens are appended to the original $19 \times 19$ grid and the combined representation is passed to the compressor for further processing.

During the video-instruction tuning stage, we uniformly sample up to 48 frames per video. Each frame is resized to $3 \times 384 \times 384$ and passed through the vision encoder, producing a $27 \times 27$ feature map. These feature maps are bilinearly pooled with a $2 \times 2$ pooling size, resulting in a $14 \times 14$ grid. To improve efficiency, we apply SlowFast-style pooling (Zhang et al., 2024c) with a stride of 3 - retaining one frame as a "slow" frame (keeping the $14 \times 14$ grid) and two as "fast" frames (further pooled to a $7 \times 7$ grid) - to produce interleaved frame embeddings. These interleaved features are passed to the clue generator, while the compressor processes the original per-frame vision features independently. Summary tokens and answer clue tokens are added in the same way as in the image-instruction tuning stage, except that the inputs are full video frames rather than image blocks.

For Stage 1 (modality alignment), we use a batch size of 256 and a learning rate of $1 \times 10^{-3}$. For Stages 2 and 3, we use a batch size of 128 and a learning rate of $2 \times 10^{-5}$. The number of learnable answer clue tokens and summary tokens are both set to 64 during training and inference. Across all stages, we use a cosine learning rate scheduler with 3% warm-up. Each stage is trained for 1 epoch on a single node with 4 NVIDIA A100 GPUs. The total training time across stages is approximately 6 days. To reduce memory usage during training, we use FlashAttention-2 (Dao, 2024) and ZeRO-2 (Rajbhandari et al., 2019), which optimize gradient, activation, and optimizer state memory.

In the MLVU (Zhou et al., 2024) dataset, the subtask "order" refers to the answer choices directly, while in the LongVideoBench (Wu et al., 2024) dataset, those subtasks are "SSS" (Sequences of Scenes), "S2O" (Scene-referred Object), and "SOS" (Scene-referred Object Tracking). In such subtasks, we always kept the answer choices attached to the query for fair comparison.

## A.2  ADDITIONAL ANALYSIS

**Compressor Ablation** The MLLM decoder is originally paired with an MLP projector that transforms vision encoder outputs before decoding. Since our $ClueSLM$ is aligned with the same decoder, it is plausible that the $ClueSLM$ may also be compatible with this original projection. To test this, we bypass the compressor and instead use the original MLP projector to transform vision features. We then extract answer clue tokens from the $ClueSLM$ and compute their relevance to the projected vision encoder features using cosine similarity between the average of answer clue tokens and the average of per-frame feature embeddings. This setup allows us to evaluate the role of the compressor. As shown in Table 5, removing the compressor leads to a noticeable drop in performance, highlighting its importance in generating high-quality clue-based relevance scores.

| Method | 8 | 16 | 32 | Avg |
|---|---|---|---|---|
| *MLVU Dev* | | | | |
| $u$ | 59.00 | 61.91 | 65.57 | 61.83 |
| $q$ | 65.09 | 67.88 | 69.77 | 67.58 |
| $q \oplus c$ (w/o comp) | 65.14 | 67.45 | 69.34 | 67.31 |
| $q \oplus c$ (w comp) | **66.01** | **68.03** | **70.00** | **68.01** |

Figure 4: Performance on MLVU (dev set) with LLaVA-Video 7B under different frame settings. $u$ - uniform sampling, $q$ - query-based selection. $q \oplus c$ (w/o comp) - compressor is excluded and $q \oplus c$ (w comp) - compressor is included, when generating clue-based frame score distribution.

**Frame & Text Encoder Ablation**    We primarily use the SigLIP-SO400M/14 model (Zhai et al., 2023) to generate the query-based relevance distribution, as its vision encoder is integrated with our clue generator during both training and inference. This setup also allows us to efficiently reuse its corresponding text encoder. As an additional ablation, we replace SigLIP with the CLIP-L/14 model (Radford et al., 2021) to generate the query-based scores and report the results in Table 3. In most cases, our clue-enhanced retrieval continues to outperform the standard query-only retrieval, demonstrating that our method generalizes across different vision-text encoder backbones.

| Frame Count | Video-MME (wo-subs) | | | LongVideoBench | MLVU |
|---|---|---|---|---|---|
| | Short | Medium | Long | val | test |
| $8_u$ | 67.7 | 53.7 | 46.9 | 55.7 | 41.1 |
| $8_q$ | 66.4 | 54.4 | 49.2 | 58.3 | **54.3** |
| $8_{q\oplus c}$ | **67.9** | **56.6** | **50.2** | **58.9** | 52.4 |
| $16_u$ | **70.9** | **58.8** | 49.9 | 56.8 | 46.0 |
| $16_q$ | 70.3 | 57.2 | 51.7 | 58.1 | 56.8 |
| $16_{q\oplus c}$ | **70.9** | 57.9 | **53.7** | **58.3** | **57.4** |
| $32_u$ | **75.9** | 59.6 | 51.6 | 57.9 | 50.4 |
| $32_q$ | 73.2 | **61.4** | 52.6 | 59.3 | 55.6 |
| $32_{q\oplus c}$ | 74.9 | **61.4** | **52.9** | **59.4** | **57.1** |

Table 3: Performance on long-form VideoQA Benchmarks with LLaVA-Video (Yu et al., 2024) using CLIP-L/14 (Radford et al., 2021) model for query-base distribution generation. FC and LVB represent the frame count and LongVideoBench, respectively. The frame count subscripts represent the sampling method: $u$ - uniform sampling, $q$ - query-based selection, where $k$ highest scoring frames are selected based on query relevance, and $q \oplus c$ - based on our proposed framework, where $k$ highest scoring frames are selected based on relevance to the query and generated answer clues.

**Merge Weight Analysis**    Using MLVU (Zhou et al., 2024) dev set, we perform analysis on the weights $w_q$ and $w_c$, which represent the weight allocated to the query-based distribution and clue-based distribution during the merging process, respectively. Additionally, we perform such analysis on the explicit answer options. We encode the query and answer options separately using SigLIP text encoder and then determine the query-based and "answer option" based distributions. In Figure 5, the best performance is achieved with $w_q = 0.7$ and $w_c = 0.3$ for the answer option case and answer clue (with *ClueSLM*) case. Both cases have a similar trend, and it can be seen that the query-based distribution carries more importance while the distribution based on explicit answer options or generated answer clue acts as a complimentary source.

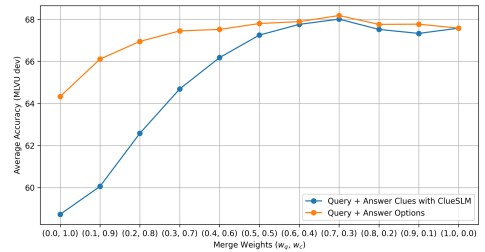

Figure 5: Effect of merge weights on frame retrieval performance using MLVU (dev) (Zhou et al., 2024) dataset.

**Broader Applicability of ClueVQA Framework**    We showcase a different way to generate answer clues as auxiliary signals in our framework. As we did before, we uniformly sample 128 frames and then pass them to a small pretrained MLLM with long-context ability, such as InternVL2.5-1B (Chen et al., 2024a), with the given query and the following additional instruction: "*Briefly describe information such as entities, objects or events that are necessary to answer this question.*" We extract the output response, encode it through the SigLIP (Zhai et al., 2023) text-encoder, and generate the clue-based distribution, which can also be merged with the query-based distribution. The optimal merge weights for $w_q$ and $w_c$ are 0.6 and 0.4, respectively, based on testing with MLVU (Zhou et al., 2024) dev set. In Table 4, we test this method under our framework and obtain considerable improvement in performance compared to the query-based retrieval. We also experiment with a different and slightly bigger MLLM, Qwen2.5-VL-3B (Bai et al., 2025a). The optimal $w_q$ and $w_c$ merge weights for this model were 0.7 and 0.3, respectively, and it showed even better results in most

| Frame Count | Video-MME (wo-subs) | | | LongVideoBench (val) | MLVU (test) |
|:---:|:---:|:---:|:---:|:---:|:---:|
| | Short | Medium | Long | | |
| $8_u$ | 67.7 | 53.7 | 46.9 | 55.7 | 41.1 |
| $8_q$ | 66.8 | 54.1 | 49.1 | 57.6 | 49.1 |
| $8_{q \oplus c'}$ | **70.3** | 55.7 | 49.8 | 58.1 | **52.4** |
| $8_{q \oplus c''}$ | 69.6 | **56.9** | **50.2** | **58.3** | 50.7 |
| $16_u$ | 70.9 | 58.8 | 49.9 | 56.8 | 46.0 |
| $16_q$ | 71.3 | 55.9 | 50.8 | 59.4 | 52.6 |
| $16_{q \oplus c'}$ | 72.8 | 57.2 | 50.8 | 59.8 | **55.6** |
| $16_{q \oplus c''}$ | **73.2** | **59.6** | **51.9** | **60.0** | 55.3 |
| $32_u$ | **75.9** | 59.6 | 51.6 | 57.9 | 50.4 |
| $32_q$ | 73.2 | 59.2 | 51.7 | 59.0 | 55.6 |
| $32_{q \oplus c'}$ | 73.7 | **61.1** | 51.8 | 59.4 | **57.8** |
| $32_{q \oplus c''}$ | 75.3 | 60.0 | **52.0** | 59.5 | 55.6 |

Table 4: Performance on long-form VideoQA benchmarks with LLaVA-Video 7B. $u$ - uniform sampling, $q$ - query-based selection, while $q \oplus c'$ and $q \oplus c''$ - based on our proposed framework, where answer clues are generated with InternVL2.5-1B (Chen et al., 2024a) and Qwen2.5-VL-3B (Bai et al., 2025a), respectively.

| Method | 8 | 16 | 32 | Avg |
|:---|:---:|:---:|:---:|:---:|
| *MLVU Dev* | | | | |
| $q \oplus c$ (harmonic mean) | 61.82 | 64.95 | 65.72 | 64.16 |
| $q \oplus c$ (arithmetic mean) | 65.08 | 67.23 | 68.59 | 66.97 |
| $q \oplus c$ (max) | 65.94 | 67.80 | 69.51 | 67.75 |
| $q \oplus c$ (weighted noisy-OR) | **66.01** | **68.03** | **70.00** | **68.01** |

Table 5: Performance of different merging methods on MLVU (dev set) with LLaVA-Video 7B under different frame settings.

cases compared to InternVL2.5-1B model. As we can see, there can be different ways to generate answer clues in our framework, while the method with $ClueSLM$ offers a more general solution with a pure small language model as a basis.

| Model | Frame Count | Video-MME (wo-subs) | | | LongVideoBench val | MLVU test |
|---|---|---|---|---|---|---|
| | | Short | Medium | Long | | |
| LLaVA-Video | $8_u$ | 67.7 | 53.7 | 46.9 | 55.7 | 41.1 |
| LLaVA-Video | $8_q$ | 66.8 | 54.1 | 49.1 | 57.6 | 49.1 |
| LLaVA-Video | $8_{q \oplus c}$ | **69.4** | **54.9** | **50.0** | **57.7** | **50.3** |
| LLaVA-Video | $16_u$ | 70.9 | **58.8** | 49.9 | 56.8 | 46.0 |
| LLaVA-Video | $16_q$ | **71.3** | 55.9 | 50.8 | 59.4 | 52.6 |
| LLaVA-Video | $16_{q \oplus c}$ | 71.1 | 56.4 | **52.0** | **59.8** | **54.9** |
| LLaVA-Video | $32_u$ | **75.9** | **59.6** | 51.6 | 57.9 | 50.4 |
| LLaVA-Video | $32_q$ | 73.2 | 59.2 | 51.7 | 59.0 | 55.6 |
| LLaVA-Video | $32_{q \oplus c}$ | 73.8 | 58.6 | **51.9** | **59.6** | **56.3** |
| InternVideo2.5 | $8_u$ | 62.8 | 52.3 | 45.9 | 50.6 | 42.3 |
| InternVideo2.5 | $8_q$ | 64.4 | 53.1 | 47.7 | 55.4 | 47.8 |
| InternVideo2.5 | $8_{q \oplus c}$ | **64.6** | **53.4** | **48.2** | **55.5** | **48.3** |
| InternVideo2.5 | $16_u$ | 65.8 | 55.7 | 47.2 | 53.6 | 43.4 |
| InternVideo2.5 | $16_q$ | **68.0** | 54.8 | 48.0 | 56.3 | 51.9 |
| InternVideo2.5 | $16_{q \oplus c}$ | 67.7 | **56.0** | **49.1** | **57.2** | **53.4** |
| InternVideo2.5 | $32_u$ | 70.3 | 57.8 | 49.7 | 55.1 | 46.3 |
| InternVideo2.5 | $32_q$ | **70.9** | **58.1** | 50.3 | **56.8** | 52.6 |
| InternVideo2.5 | $32_{q \oplus c}$ | 69.9 | 57.9 | **51.6** | 56.7 | **54.1** |
| Qwen2.5-VL | $8_u$ | 62.1 | 51.2 | 47.0 | 52.5 | 37.0 |
| Qwen2.5-VL | $8_q$ | 63.2 | 52.8 | 47.9 | 55.8 | 44.5 |
| Qwen2.5-VL | $8_{q \oplus c}$ | **65.7** | **53.2** | **50.1** | **57.1** | **45.2** |
| Qwen2.5-VL | $16_u$ | 66.2 | 56.2 | 48.3 | 55.2 | 38.9 |
| Qwen2.5-VL | $16_q$ | 65.4 | 56.7 | 51.9 | 57.7 | **46.5** |
| Qwen2.5-VL | $16_{q \oplus c}$ | **68.3** | **56.8** | **52.2** | **58.0** | 46.3 |
| Qwen2.5-VL | $32_u$ | **71.7** | 59.3 | 50.6 | 58.7 | 42.2 |
| Qwen2.5-VL | $32_q$ | 69.4 | **60.3** | 51.3 | 58.9 | 47.3 |
| Qwen2.5-VL | $32_{q \oplus c}$ | 71.3 | 58.8 | **52.9** | **59.1** | **47.9** |

Table 6: Performance on long-form VideoQA benchmarks with LLaVA-Video (Yu et al., 2024), InternVideo2.5 (Wang et al., 2025), and Qwen-2.5 VL (Bai et al., 2025a) using SigLIP (Zhai et al., 2023) model for query-based distribution generation. All 3 Video-LLMs are approximately 7B in size. Bolded score indicates the highest performance, underlined score indicates the second highest performance. The frame count subscripts represent the sampling method: $u$ - uniform sampling, $q$ - query-based selection, where $k$ highest scoring frames are selected based on query relevance, and $q \oplus c$ - based on our proposed framework, where $k$ highest scoring frames are selected based on relevance to the query and generated answer clues.

