# OpenReview forum: "ClueVQA: Enhancing Query Based Retrieval in Video-LLMs with Answer Clues"
_ICLR.cc/2026/Conference — ICLR 2026 Conference Withdrawn Submission_

### Official Review · Reviewer_u6hE · 2025-10-26

**Soundness:** 2
**Presentation:** 3
**Contribution:** 2
**Rating:** 2
**Confidence:** 4

**Summary:**

The paper presents a new framework to improve long-form video understanding, by incorporating a SLM-based answer clue generation that complements query-based retrieval. The system also incorporates a compressor to summarize frame features into compact tokens, reducing token load while maintaining semantic fidelity. Experiments across multiple benchmarks show consistent but modest performance improvements over query-based retrieval.

**Strengths:**

The paper directly addresses the limitation of query-based frame retrieval in long-video question answering. It’s interesting to incorporate the answer-oriented clues via a lightweight small language model, which extends retrieval beyond surface-level similarity and provides a new perspective on integrating latent reasoning cues.

The proposed scheme is designed to be model-agnostic and compatible with multiple existing Video-LLMs, showing implementation flexibility.

The paper is also well-written with clear motivation and problem definition.

**Weaknesses:**

The proposed method’s conceptual contribution is clear, yet its practical significance is undermined by missing efficiency analysis, small performance gains, and lack of a full-frame reference baseline.

The gains over query-based retrieval are marginal. This improvement might be relatively small compared to the additional model complexity and computation introduced by ClueSLM.

The paper lacks comparison with existing agent-based or reasoning-enhanced video understanding systems (e.g., VideoAgent, Goldfish, or VideoTree).

The paper hasn’t included any measurement or discussion of the added computational cost from the small language model and from encoding 128 frames per query. There is no runtime, FLOPs, or latency comparison with the baseline, leaving efficiency claims unsubstantiated.

Table 2 evaluates the compressor module, but its real impact on end-to-end retrieval or QA efficiency is unclear. It’s suggested to add more discussions.

**Questions:**

Please address the issues listed in the weakness part.

---

### Official Review · Reviewer_1gus · 2025-10-30

**Soundness:** 2
**Presentation:** 2
**Contribution:** 3
**Rating:** 4
**Confidence:** 5

**Summary:**

This paper presents ClueVQA, a retrieval-enhancement framework for long-form VideoQA with Video-LLMs. The key idea is to augment query-based frame retrieval with generated answer clues—latent, answer-oriented representations derived from the query and a global scan of the video. A compact module, ClueSLM, is trained in two modes (compression and clue generation) and produces clue-based frame relevance scores, which are fused with query-based scores via a generalized noisy-OR mechanism.

**Strengths:**

+ Motivation is clear.
+ The paper introduces the idea of latent answer clues as a complementary signal to query-based retrieval. This goes beyond standard semantic matching, addressing implicit and reasoning-heavy questions.
+ Good performance is achieved.

**Weaknesses:**

1) There is a lack of deep analysis why the clue generator can address the pinpointed challenge.  For example, for the query "Why does the driver slow down before the interaction?" could you visualize the selected frames for your method vs. simple retrieval strategy?
What is the network architecture of the clue generator?

2) The "answer clue" tokens are somewhat like the learnable tokens in Q-former (e.g., in BLIP2). Differently, those tokens are directly input to LLM for visual reasoning while your learned tokens are used to compute the correlation with frame features for frame selection. How is the performance when you simply concatenate the clue tokens and the compressed tokens into decoder (with learnable projection)? What is the advantage of your current design?

3) In Table 1, how about the performance when using c only? When clue work better and when query works and why?

**Questions:**

See weaknesses.

---

### Official Review · Reviewer_ztYG · 2025-11-01

**Soundness:** 3
**Presentation:** 3
**Contribution:** 2
**Rating:** 4
**Confidence:** 4

**Summary:**

Brief Summary: The paper tackles the task of video frame retrieval for video qa where given a question and a video the task is to find relevant frames from a long-video which are then passed to a separate VLM for answering the question. Standard approach is some variation of RAG, but here authors suggest a nice idea of using small-language model (ClueSLM) which acts to generate both a clue-generator and frame-summary tokens. Finally, the similarity between the answer scores and the frame-summary tokens provide the top-K retrieved frames. The small-language model is based on Rene which is mamba-based arch 1.3B model, training is done in three stages including alignment, image and then video instruction tuning. Experiments are conducted on standard long-video understanding benchmarks such as Video-MME, MLVU, LongVideoBench, where authors show proposed method consistently improves performance.

**Strengths:**

Pros:

1. Video frame retrieval for Video-QA is widely studied and important task. The paper is well motivated on that front.

2. The architecture idea is novel to best of my knowledge. Essentially, using the query + frames to generate embeddings for answer clues, and then using the same model with frames + summary token representation. Due to the nature of the task itself, it can be a plug and play with any model.

3. The improvements are mostly consistent (Table 1), and in Table 2 the authors show the compressor part is improving over standard adaptive pool

**Weaknesses:**

Cons:

1. My main concern is that while the improvements are present consistently, the absolute improvement appear quite marginal especially such as on video-mme long subset which is where the retrieval part should ideally shine. As such it doesn't seem there is significant value in going through this process compared to simple uniform sampling.

2. The authors have missed an obvious ablation of using separate ClueSLM models for Clue Generator and Compressor where the weights are not shared. It is also not clear what is the main advantage of having the same model for both cases??

3. There is no visualization of the types of answer generated by the clue-decoder? Also, some visualization of which frames are selected would be good, in particular for the cases where proposed method does better than baseline.

4. The authors should include some ablations where instead of the clue generator and compressor, simple off-the-shelf VLM is used which does query-rewriting followed by RAG for fairer comparison. Currently, no baseline methods are compared against with. It is a bit unclear why authors haven't compared with sevila, video tree which they mention in related works?

5. Instead of mamba based SLM, what if we used transformer based such as llama-3.2 of similar size? Are there any trade-offs?

---

Overall Rating: 4/10
While the proposed method outperforms baselines, many other baselines which are mentioned in related works itself are not directly compared against which is surprising. The core idea is similar to query-rewriting+RAG, some comparison should be made. The main reason to have coupled clueslm is not directly validated, also unclear about the motivation to go to mamba-like architecture.

**Questions:**

Q1. Is there a chance of train/test leakage? ClueSLM is trained on multiple datasets including activitynetqa, which could overlap with mlvu/longvideo-bench?

---

### Official Review · Reviewer_gYEg · 2025-11-01

**Soundness:** 3
**Presentation:** 3
**Contribution:** 2
**Rating:** 4
**Confidence:** 4

**Summary:**

This paper introduces ClueVQA, a retrieval-enhancement framework for long-form VideoQA that improves frame selection by generating and integrating supplementary answer clues. The core idea is to move beyond simple query-frame similarity by training a small language model (ClueSLM) to operate in dual modes: as a frame compressor and as a clue generator that produces latent answer clues from a global video scan. These clues create a secondary relevance distribution that is fused with the standard query-based distribution via a generalized noisy-OR mechanism. The method demonstrates consistent improvements over query-only retrieval across multiple Video-LLMs and long-form VideoQA benchmarks (VideoMME, LongVideoBench, MLVU), while maintaining compatibility with existing models.

**Strengths:**

1. The idea of generating implicit answer clues to complement query-based retrieval is innovative and addresses a clear limitation of current methods. The preliminary analysis showing performance gains when using ground-truth answer options provides strong motivation for the approach.

2. The framework is well-designed and modular, making it compatible with various existing Video-LLMs and retrieval backbones without requiring architectural changes. The dual-role ClueSLM (compressor + clue generator) is a clever design that enables efficient computation in a shared latent space.

3. The paper provides extensive experiments across three challenging long-form VideoQA benchmarks and multiple Video-LLMs (LLaVA-Video, InternVideo2.5, Qwen2.5-VL). The consistent improvements across different frame budgets (8, 16, 32) and models demonstrate the robustness of the approach.

4. The paper includes valuable ablation studies on merging strategies, compressor importance, different vision-text encoders, and alternative clue generation methods. The t-SNE analysis showing that latent clues form meaningful clusters by video category provides interpretability and validates that the clues encode semantically relevant information.

**Weaknesses:**

1. While the paper shows improvements over basic query-based retrieval, it lacks comparison to more sophisticated retrieval approaches like VideoTree, SALOVA, or other recently learned retrieval methods. This makes it challenging to evaluate how ClueVQA compares to the current state-of-the-art in video retrieval.

2. The query-only baseline uses simple cosine similarity between query and frame embeddings. More advanced query-based methods (e.g., those incorporating cross-attention or learned scoring) may narrow the performance gap, making the improvements appear less significant.

3. While the method uses a relatively small ClueSLM, the whole pipeline requires: (1) dense frame sampling and feature extraction, (2) ClueSLM inference in two modes, and (3) distribution fusion. The paper doesn't provide a comprehensive analysis of the additional latency and computational cost compared to standard retrieval.

4. The paper doesn't sufficiently analyze when the method fails or under what conditions clue-based retrieval might perform worse than query-only. Understanding the limitations (e.g., for very ambiguous queries or when the clue generator hallucinates) would strengthen the contribution.

**Questions:**

See the weakness part.

---

### Note · Authors · 2025-11-13

I have read and agree with the venue's withdrawal policy on behalf of myself and my co-authors.